# The Impact of the Technological Capability of a Host Country on Inward FDI in OECD Countries: The Moderating Roles of Institutional Quality

**Seunghyun Kim [1] and Byungchul Choi [2],***

[1]   Science and Technology Policy Institute, Sejong 30147, Korea; shkim@stepi.re.kr
[2]   College of Business, Hankuk University of Foreign Studies, Seoul Campus, Seoul 02450, Korea
*   Correspondence: bchoi@hufs.ac.kr; Tel.: +82-2-2173-3090

**Abstract:** This empirical study explores the impacts of technological capability on inward foreign direct investment (FDI) with the moderations of institutional quality. We extend the existing literature by contributing the dynamic links between technology trade and institutional quality by using the panel data of 35 Organization for Economic Cooperation and Development (OECD) countries between 2000 and 2015. Based on fixed-effects regression, our results show that there is a U-shape relationship between the net technological capability of a host country and inward FDI. In addition, the institutional quality of a host country, government size and regulation have positive moderations, whereas sound money accessibility and legal system and property protection have negative moderations on the main U-shape relationship. Our study contributes to the literature on the determinants of inward FDI in the context of technological capabilities and institutional quality.

**Keywords:** foreign direct investment; technological capability of a host country; institutional quality

---

## 1. Introduction

Foreign direct investment (FDI) is one of the main driving forces behind cross-border technology spillovers [1,2] and is preferred by developing countries due to its positive influence on economic development [3]. In this sense, previous studies have focused on the determinants of FDI inflows to developing countries and have highlighted the importance of institutional quality in host countries in attracting FDI [2–4]. On one hand, high institutional quality leads to high inflows of FDI and low volatility of FDI [5], while established property rights also have a positive influence on FDI inflows [6–8]. On the other hand, there are negative determinants of FDI inflows, such as corruption [9] and institutional distance between the home and host countries [10]. While these studies have highlighted institutional quality as a determinant of FDI, one commonly acknowledged premise is the direct relationship between institutional quality and FDI; a potential direct link between institutional quality and FDI has been shown. However, institutional quality sets "the rules of the game" at the macro level; its influence may work as a secondary rather than a primary determinant of FDI. As the ownership, location, and internalization (OLI) framework illustrates ([11,12]), there are primary motives for investors to allocate their resources to other countries.

Recognizing this gap, the present study investigates the association between the technological capability of host countries and inward FDIs. While previous studies have focused on the technological spillover effects through FDI from developed countries to developing countries or vice versa [11–14], a host country's technology capabilities as a determent of inward FDI has received relatively scant attention from researchers, especially in the context of inward FDI to developed countries. By using the notion of technology balance of payments (TBP) and inward FDI data from OECD countries, we argue

that there is a curvilinear relationship (U-shaped) between the inward FDI and the technological capability of a host country. TBP consists of four categories: transfer of techniques (this includes patents, licenses, disclosure of know-how and transfer of designs), trademarks and patterns, transfer of services with technical content and industrial R&D. These can estimate the flow of technological assets of a country [15]. We focus our attention on how the various aspects of the institutional quality of a host country moderate this U-shaped relationship between inward FDI and the technological capability of a host country. Four dimensions of institutional quality are employed: the size of government, legal structure and security of property rights, access to sound money and regulation of credit, labor and business.

This study is exploratory research to examine the relationship between TBP and FDI. It aims to suggest a complementary variable that can reflect the technology flow and related firm's activity aspect as well as technology competitiveness of technological capability and provide a basis to extend the scope of understanding of the relationship between the technological capability of a nation, institutional quality, and inward FDI. We examined 35 OECD countries between 2000 and 2015 and found that there was a U-shape relationship between the net technology trade and inward FDI. In addition, government size and regulation had positive moderations, whereas sound money and legal system and property protection had negative moderations on the main U-shaped relationship.

This paper is structured as follows: In the next section (Section 2), the conceptual framework is established through a review of the literature. Section 3 presents the empirical research setting. Section 4 shows the empirical results. Section 5 is a discussion of the main results. Lastly, Section 6 concludes and points out future lines of research and the paper's limitations.

## 2. Theoretical Framework

Dunning [11,12] proposes the eclectic paradigm, also known as the ownership, location and internalization (OLI ) framework, for the determinants of FDI. Ownership advantages explain the motivations for FDI—investors gain more ownership when they expect higher competitive advantages in foreign production. Location advantages are exploited when investing firms expect value-adding activities. This advantage is particularly preferable when the resources are immobile, natural or created. Internalization advantages are desirable when firms want to create or exploit core competencies. In the OLI model, FDI to achieve technological capabilities is primarily based on the location and internalization advantages. In other words, the level of FDI will vary depending on the nature of the host countries. For instance, Palit and Nawani [13] suggest that one of the reasons East Asian countries such as Korea, Singapore and Hong Kong are attracting more FDI than other Asian countries is their distinguished technological capabilities.

Biggs et al. [16] define "technological capabilities" as the information and skills in terms of technical, managerial and institutional for productive enterprises to utilize equipment and technology efficiently. In this regard, technological capabilities include a wide range of firm's activities, in-house technologies itself, imported technologies, buying skills and firm strategies. Archibugi and Coco [17] suggest three main components of technological capabilities: the creation of technology, technological infrastructures and the development of human skills. Previous studies show various factors affecting technological capability changes. The exploitation of external knowledge including licensing, patent, R&D expenditure, infrastructures including Internet penetration, electricity consumption, mean years of schooling, literacy rate are used as variables for technological capability [13,17,18].

The importance of knowledge management has been well documented in the international business (IB) literature [19–21]. In this stream of literature, searching for technology capabilities is considered as one of the essential motivations for a firm's pursuit of FDI. There are many previous studies that have focused on technological capabilities as primary determinants of FDI. Bah et al. [14] suggest that FDI is an effective instrument for transferring needed technologies and knowledge to a host country when investors in the home country expand their marketplace. Sharma and Bandara [22] use the knowledge capital variable to explain the motivations for FDI among developed countries.

They adopt technology-level differences to explain the different levels of FDI from Australia. China is one of the popular countries in recent literature used to explain the dynamics of the technology spillover effects by FDI. Liu and Guo [23] show the evolutionary technology spillover through FDI using data for the period 2003–2014. Liu et al. [24] focus on renewable energy technology spillover through FDI. Investigating this issue is becoming more important and interesting since, over the last two decades, emerging market multinational enterprises (EMNEs) has begun to play a more active role in technology-intensive industries [25,26], and FDI has been viewed as a key instrument of learning knowledge from developed economies [27,28].

Although many scholars have focused on the technological capabilities through FDI and found a significant relationship with FDI, few studies have investigated the effect host country's TBP on FDI. Considering the definition of technological capability, TBP is also appropriate as a variable to represent it. Import of technologies means the exploitation of external knowledge. Avallone and Chédor's [29] research also shows the link between R&D and TBP. FDI is not just from the developed countries to underdeveloped countries to access the knowledge needed. The goals of investing capital either in developed or underdeveloped countries are different and are similar to the problemistic search and slack search model suggested by Cyert and March [30]. Problemistic search refers to a situation when firms seek ways to enhance existing operations to achieve a target, while the slack search is for facilitating a firm's adaptation [31]. When investing firms engage in FDI in underdeveloped countries, the primary motivation is to acquire local knowledge to modify their existing products or services, as suggested by the problemistic search part of the model. However, when firms engage in FDI in developed countries, they seek to acquire unexploited knowledge, available only in the developed countries, as suggested by the slack search part of the model [32]. Although there exist clear motivational distinctions between FDI in developed and underdeveloped countries, there is a lack of literature pertaining to the dynamics of FDI motivations.

Besides the technological level of a country, FDI may vary depending on the institutional quality. After North's [33] pioneering work, institutions are considered one of the essential components for explaining the differences between countries' and firms' strategic decisions. Institutions define the "rules of the game". From a different perspective, North [33] defines an organization as " … purposive entities designed by creators to maximize wealth, income, or other objectives defined by the opportunities afforded by the institutional structure of the society". However, institutions are not static; they continue to change through market reforms from a government-led economy to a market-led economy [34]. In other words, a company makes different strategic choices (i.e., R&D investment) as institutional settings change [34,35]. Few studies in FDI literature have focused on FDI decisions incorporating institutional quality. For instance, Sharma and Bandara [22] empirically test the FDI decision from Australia to other countries and find a preference to invest in countries with high economic stability and institutional credibility. Coe, Helpman and Hoffmaister [2] also consider institutions, such as ease of doing business, education quality, patent protection and legal system in terms of international R&D spillovers. To address the argument discussed above, we explore the determinants of inward FDI by exploring the dynamics between technological capability and institutional qualities of host countries. Figure 1 illustrates the conceptual model of our study.

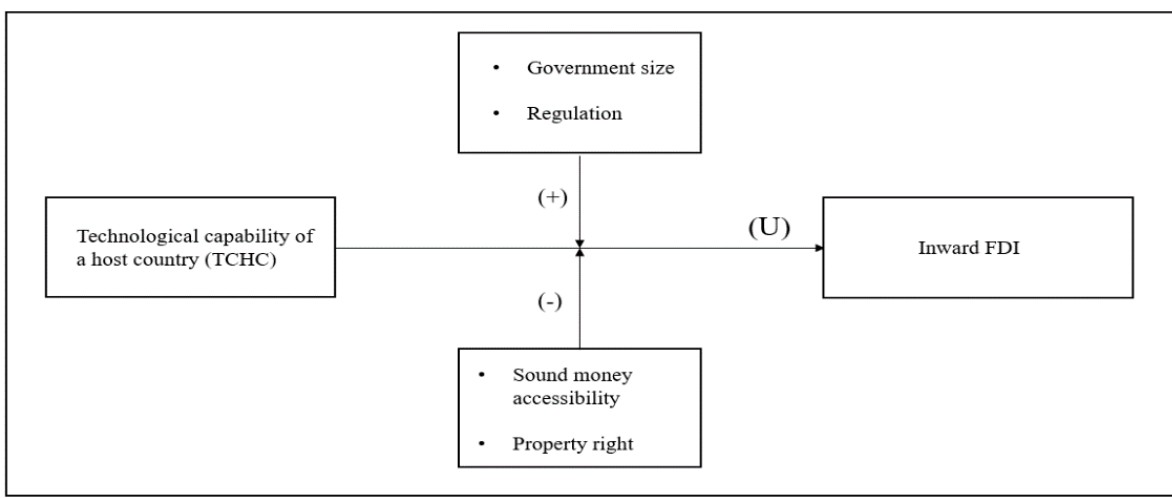

**Figure 1.** Conceptual model.

## 3. Hypothesis Development

### 3.1. Technology Capabilities of a Host Country and Inward FDI

When the technological capability of a host country is less competitive, FDI can be a desirable option for both investors and investees. From the perspective of investor companies, while the internal expansion of markets is one of the key strategies for the growth of the firm, market diversification activities lacking a thorough understanding of the target country often lead to devastating outcomes, such as the failure of Walmart's entry into the Korean market [36]. Due to the liabilities of foreignness, investors tend to increase knowledge with the aim of accessing strategic resources in the target country (known as "resource-seeking" FDI) [37] and extending the international channels for marketing products (known as "marketing-seeking" FDI) [22,38]. From the perspective of investees and their home countries' government, FDI is beneficial: multinational enterprises (MNEs) deploy their investments to access the assets of developed countries, such as technology, management skills, product design, quality characteristics and others [38]. The spillovers of such capabilities are crucial assets for economic growth through nurturing domestic entrepreneurship and entrepreneurial talent.

On the other hand, FDI also inflows to technologically developed countries or among developed countries based on a specific need for knowledge. From the perspective of investor firms, accessing advanced technologies, managerial skills, and highly educated knowledge, workers can be major motivations for FDI from developing countries to developed countries. Recently, determinants such as new knowledge capital have been added to explain FDI [39,40]. Similarly, Cantwell [41] posits that cross-border technological variations are the main motivations for MNEs. For instance, " ... both Japanese and European multinationals are setting up R&D units in each other's markets with a view to responding to local customers' needs and tastes, as well as to capturing the locality-specific innovation in order to produce new products and varieties which cannot only be sold locally but also exported to other countries" [42].

Hence, rather than greater FDI inflows to countries with mediocre technologies, investing firms tend to bipolarize their investment strategies into either marketing-seeking FDI or technology-seeking FDI, based on the level of technological capability of a host country. By combining the arguments presented above, we hypothesize:

**H1.** *There will be a* U*-shaped relationship between inward FDI and the technological capability of a host country.*

### 3.2. Moderations of Institutional Quality

Institutions critically matter for an organization's strategic decision. The theory of institution and institutional change by North [33] provides a theoretical background to understanding changes of environments such as formal rules, structuring incentives and constraints and informal norms and conventions form and their impact on firms' behavior. Since the seminal contribution of North [33], numerous studies in diverse disciplines including international business (IB) and strategic management have well documented that the institutional nature of the country significantly influences a firm's strategic choice and economic activities of stakeholders [43–48]. In this stream of literature, one of widely acknowledged argument is that intuition can have two faces—a supporter that helps firms resolve issues that a single firm cannot handle or a control tower that constrain a firm's strategic intention [49,50].

Literature in this stream argues that the institutional matrix with formal policies or rules governs economic activities. The institutional matrix represents different formal rules; for example, the formal rules in a socialistic economy are usually controlled by the government, which plays a central role in monitoring and enforcing the formal rules. On the other hand, the formal rules in a market economy reside in the market forces pursuing profit maximization of its entities. Pro-market reform is a process of changing formal rules from the government to market forces. Existing work on pro-market reforms in emerging economies has primarily focused on the influence of reforms on firm performance, with contradictory results. On one hand, increased external monitoring influences firm efficiency, raising performance [51]. Additionally, firms benefit from increases in growth opportunities and access to resources that were out of their reach [52,53] before market reforms. On the other hand, increased competition from new entrants leads to reduced firm profitability [54]. While in the long run, increased competition is beneficial for product efficiency and innovation, as the process of market reform unfolds, established firms that are used to stable markets and government protection may face decreased performance due to increased competition. Although a few studies have focused on heterogeneous FDI levels based on institutional quality [2–4], little study has been done in terms of the technological capability aspects of FDI.

As mentioned above, the nature of the institution can be either supportive or oppressive for a firm's strategic decision, such as investments. By recognizing these two faces of an institution, we explore the diverse contingencies by moderating institutional quality measures: the size of government, legal structure and security of property rights, access to sound money and regulation of credit, labor and business. We argue that these four institutional-quality factors may critically influence the technological assets-related overseas investment of the firm. Technology, by its own nature, is generally intangible assets, expensive to explore and exploit and is not free from the risk of information asymmetry between investors and investees. Therefore, we assume that the degree of government intervention (size of government), financing and maintaining the value of assets (access to sound money), protecting knowledge (legal structure and security of property rights) and business environment (regulation of credit, labor and business) are all salient institutional quality to investors (of inward FDI).

#### 3.2.1. Positive Moderations

Size of government and the regulation of credit, labor and business qualities, these two institutional variables (e.g., the government's tax and spending, the level of freedom and regulation of market activities, etc.) may be viewed as the economic and political distance between the investor and the investee. Berry et al. [55] measured cross-national institutional differences in terms of multidimensional variables, including economic, financial, political and administrative distance and also verified the significance of their correlation [56]. In the case of Hines Jr and Rice [57], a 1% drop in the local tax rate in the investee country increased the facility assets of the foreign investor by 3%. In other words, regulations, taxes and governmental policies that affect the actual profit of a company can increase or decrease investment; these effects can be matched with technological capabilities to accelerate or

decelerate investment. Therefore, in this study, the hypothesis that the above two variables would have a positive effect was established. The size of government implies " . . . countries with lower levels of government spending, lower marginal tax rates, and less government investment and state ownership assets" [58].

In the case of low technology-capability countries, the role of government is essential to initiate or operate businesses properly, especially for foreign-invested companies [34]. If there is little support from the host country's government, firms may face many obstacles in conditions of unestablished market constituencies. Hence, the negative slope of the low technological capability against the inward FDI curve becomes steeper when the host country has a high size of government score, meaning less intervention by the government (In our data source, the size of government is measured with a reversed score). However, the positive slope of the high technological capability against the inward FDI curve is also steeper with a high size of government score. This is because these countries already have an established market system, along with high technological capabilities. Rather than a highly regulated one, a market-oriented system is preferable for foreign investors seeking higher or quicker returns on their investment.

The same reasoning applies to regulation. Regulation is measured as " . . . how regulations that restrict entry into markets and interfere with the freedom to engage in voluntary exchange reduce economic freedom." When countries have low technological capabilities, the motivation for investing in foreign companies is not primarily to access technologies but to access markets. High scores of regulation play a significant role in drawing much attention from foreign investors seeking to expand markets. Hence an established regulation (less intervention) leads to higher inward FDI. In the case of countries with high technological capabilities, invested companies can appropriate their technologies under highly protected environments, also leading to higher inward FDI. Hence, we hypothesize as follows:

**H2.** *Size of government positively moderates the* U-shape*d relationship between the technological capability of a host country and inward FDI.*

**H3.** *Regulation positively moderates the* U-shape*d relationship between the technological capability of a host country and inward FDI.*

3.2.2. Negative Moderation

Although many institutional qualities enhance inward FDI, especially when countries are at the extreme ends of the technological capability spectrum, there is an institutional quality that works in the opposite direction. Sound money means " . . . money with relatively stable purchasing power across time—reduces transaction costs and facilitates exchange, thereby promoting economic freedom" [58]. When countries have low technological capabilities, access to sound money means lower transaction costs in the financial markets, and thus the negative slope becomes less steep. It is because the ability to access sound money can be a buffer for invested firms. However, it also increases slowly as technological capabilities increase on the positive side of the relationship between technological capability and inward FDI. Investing firms seek technology rather than financial resources. Accessing sound money is insignificant in the case of foreign investors seeking new technological capabilities.

Legal system and property rights means " . . . rule of law, security of property rights, an independent and unbiased judiciary, and impartial and effective enforcement of the law" [58]. These two variables can be considered from the perspective of the ownership advantage approach by the investor companies. According to the OLI paradigm, an overseas corporate investment, FDI, is one of the means to maximize its monopoly rent [59]; therefore, the stability of the exchange rate or inflation, intellectual property (IP) protection and other legal and institutional systems can be a criterion for determining whether to secure the ownership advantage for the investor. However, these conditions are a strict prerequisite for business activities, and the actual performance can be influenced by tax rates, freedom of business activities, regulations or government intervention. Therefore, these two institutional variables are

rather a motivation for investment than causes of acceleration or deceleration of FDI. Therefore, in this study, a hypothesis was established that access to sound money and a legal structure and security of property rights might have a negative impact in terms of the rate of increase or decrease in investment amount.

**H4.** *Sound money negatively moderates the* U-shape*d relationship between the technological capability of a host country and inward FDI.*

**H5.** *Legal system and property rights negatively moderate the* U-shape*d relationship between the technological capability of a host country and inward FDI.*

## 4. Data and Methodology

### 4.1. Data and Sample

In this study, we use panel data which span the period 2000–2015 and comprise 35 countries (Countries include Argentina, Australia, Austria, Belgium, Canada, Czech Republic, Denmark, Estonia, Finland, France, Germany, Greece, Hungary, Ireland, Israel, Italy, Japan, Korea, Lithuania, Mexico, Netherlands, New Zealand, Norway, Poland, Portugal, Romania, Russia, Singapore, Slovak Republic, South Africa, Spain, Sweden, Switzerland, United Kingdom, United States). The statistical data are gathered from OECD main science and technology indicators (MSTI) DB [60], OECD MEI(Main Economic Indicators) DB [61], World bank DB [62] and Fraser Institute DB [63]. FDI net inflow and Total GDP data are gathered from the World bank DB. Data for trade in goods, i.e., international trade exports and imports, are collected from OECD MEI DB and technology trade balance, receipts, and payments of TBP data are from OECD MSTI DB. Countries' institutional quality data are from the "Economic Freedom of the World"' report of the Fraser Institute [58] (https://www.fraserinstitute.org/). They measure the market freedom index using five categories (see Appendix A). Our final sample comprises 489 country-year observations for the period 2000–2015. We use 2000 as the cutoff year because the institutional quality index provides annual estimates of the factors only after 1999.

### 4.2. Variables

*Dependent variable: Inward FDI* (FDI net inflow) refers to the capital from foreign investors participating in the management activities of the recipient country. It means the balance of investment retrieval from new investment inflows. FDI data (urrent US dollars) used in this study are estimated by the World bank from the UNCTAD and official national sources. The definition of FDI net inflow is based on the Balance of Payments Manual 6th edition from International Monetary Fund (IMF) [64] and consists of: equity investment, including investment associated with equity that gives rise to control or influence, investment in indirectly influenced or controlled enterprises, investment in fellow enterprises, debt (except selected debt), and reverse investment [64].

*Independent variables*: To estimate the technological capability of a host country (TCHC), we use the concept of the technology balance of payments (TBP). TBP constitutes the disembodied technology diffusion and has four categories: transfer of techniques (through patents and licenses, disclosure of know-how), transfer (sale, licensing, franchising) of designs, trademarks and patterns, transfer of services with technical content, including technical and engineering studies, as well as technical assistance and industrial R&D [15]. TBP seeks to identify all intangible transactions related to technical knowledge and services between countries, and TBP data should meet three conditions: First, it must be a transaction between different countries, that is, an international transaction. Second, as it is a commercial transaction, there must be expenditure and income between the parties. Third, the scope of the transaction should be related to the trade of technology and technology services. In this respect, TBP has characteristics of technological capability as it includes transactions in technology and know-how such as patents and licenses [29,65,66]. The technological capability of a host country is calculated by subtracting the TBP payments (the amount of money for technology imports) from the

TBP receipts (the amount of money received for exporting technology). Hence, a positive value for TBP means a surplus in technology trade, while a negative value means a deficit. Superior performance in TBP may reflect that a country commands a competitive position in technology development, indicating a country with high technological capability even among OECD countries. Teixeira and Barros [67] show that TBP surplus fosters OECD 26 countries' international competitiveness. TBP does not show one country's technology level, but a balance of technology trade. Some countries create surplus through high-tech exports, while others with high-tech record deficits. However, whether Teixeira and Barros's [67] study shows, except for specific countries, it can be said that countries with a high level of technology generally record a surplus. TBP data are extracted from national sources based on the OECD manual of the proposed standard method of compiling and interpreting technology balance of payments data [68,69].

Figure 2 illustrates TBP by country in our sample. The data used in this study cover 35 countries. Based on the technology trade balance, only five countries (Ireland, Italy, Poland, Portugal and Romania) have changed their deficit and surplus positions between 2011 and 2015. The deficit and surplus positions were maintained in 30 countries. Considering the year 2015 rankings (million dollars), for TBP receipts, that is, technology exports, the top countries were the United States (130,834), Ireland (73,337) and Germany (71,836), whereas, for TBP payments (technology imports), the order was Ireland (98,091), the United States (88,891) and Germany (53,734). For technology trade balance, the United States (+41,943), Japan (+33,472), UK (+19,780), Germany (+18,102) recorded a surplus, followed by Ireland (−24,754), Korea (−6001), Switzerland (−3662) and Australia (−3372) in deficit positions. Of these countries, the United States, the United Kingdom and Japan showed a technology trade surplus but a commodity trade deficit, while Korea, Switzerland and Ireland posted a technology trade deficit but a surplus in commodity trade. In the case of Germany, both commodity and technology trades were in surplus. Technology trade data include the flow of technological know-how and services into and out of the economy, i.e., transactions in disembodied technology. These transactions could show a country's technological capacity and future economic development potential. Therefore, they could be used as an indicator to represent determinants of foreign investment. To estimate the institutional quality of a country, we employ four institutional variables: government size, property rights (legal structure and security of property rights), sound money accessibility (access to sound money) and regulation (regulation of credit, labor and business). Each construct is measured based on multiple subconstructs (See Appendix A).

*Control variable*. Total GDP (GDP at purchasers' prices (current US dollars)) is the aggregate gross value added by all resident producers in the economy plus any product taxes, minus any subsidies not included in the value of the products [62]. It includes depreciation of fabricated assets or depletion and degradation of natural resources. Total GDP represents the current economic scale of a country. Therefore, it could explain the current level of development of an economy by supplementing the potential for future economic development (technology trade balance variable). Trade in goods (exports and imports in goods) represents the import and export of material resources of a country. Goods are physical items of which ownership rights can be established and traded. Data are gathered by the 2008 system of national accounts (SNA). Trade in goods is a representative index of international transactions, along with capital transactions such as FDI and nonembodied technology transactions such as technology trade balance. It complements the explanatory power of independent variables in terms of tangible item transactions.

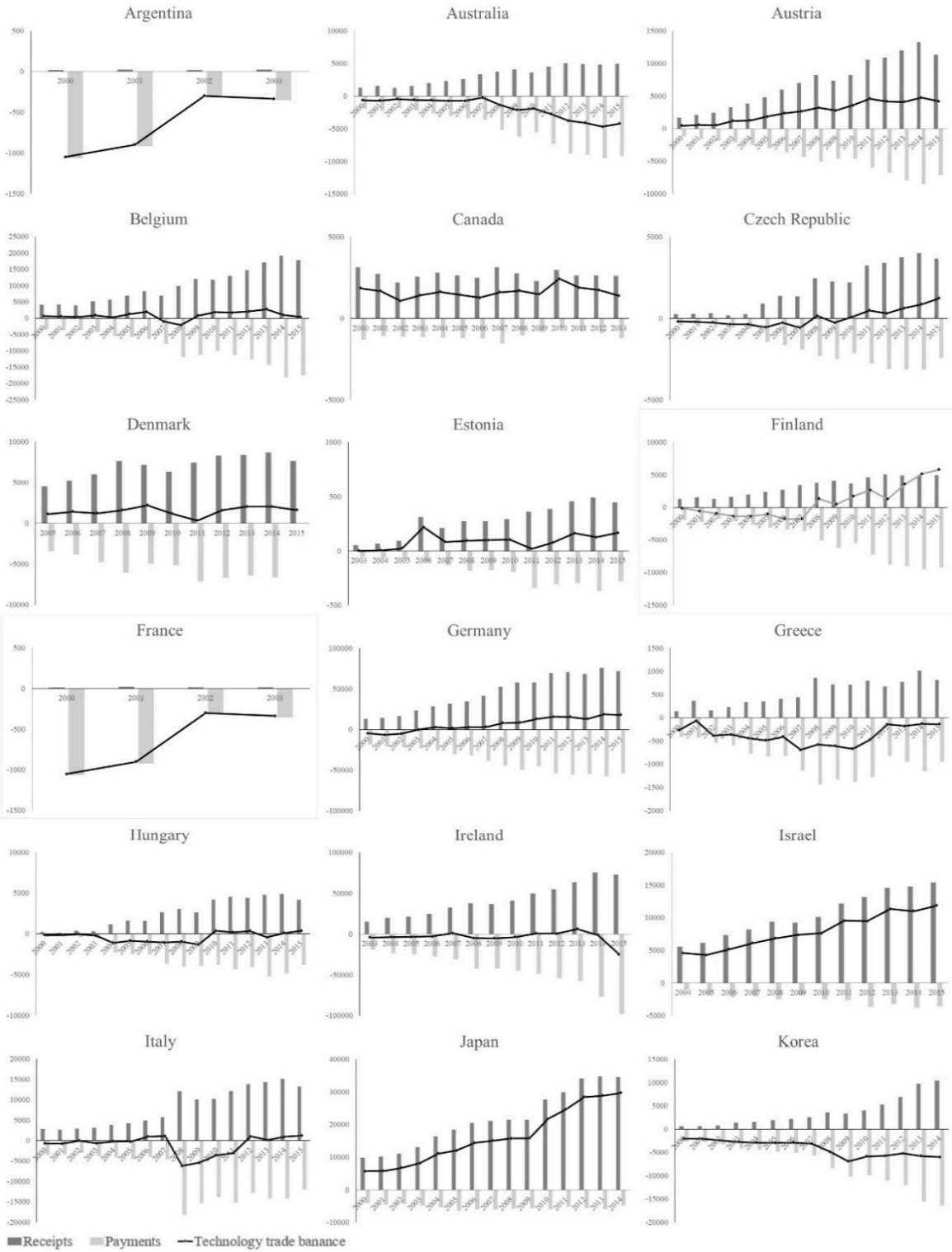

**Figure 2.** *Cont.*



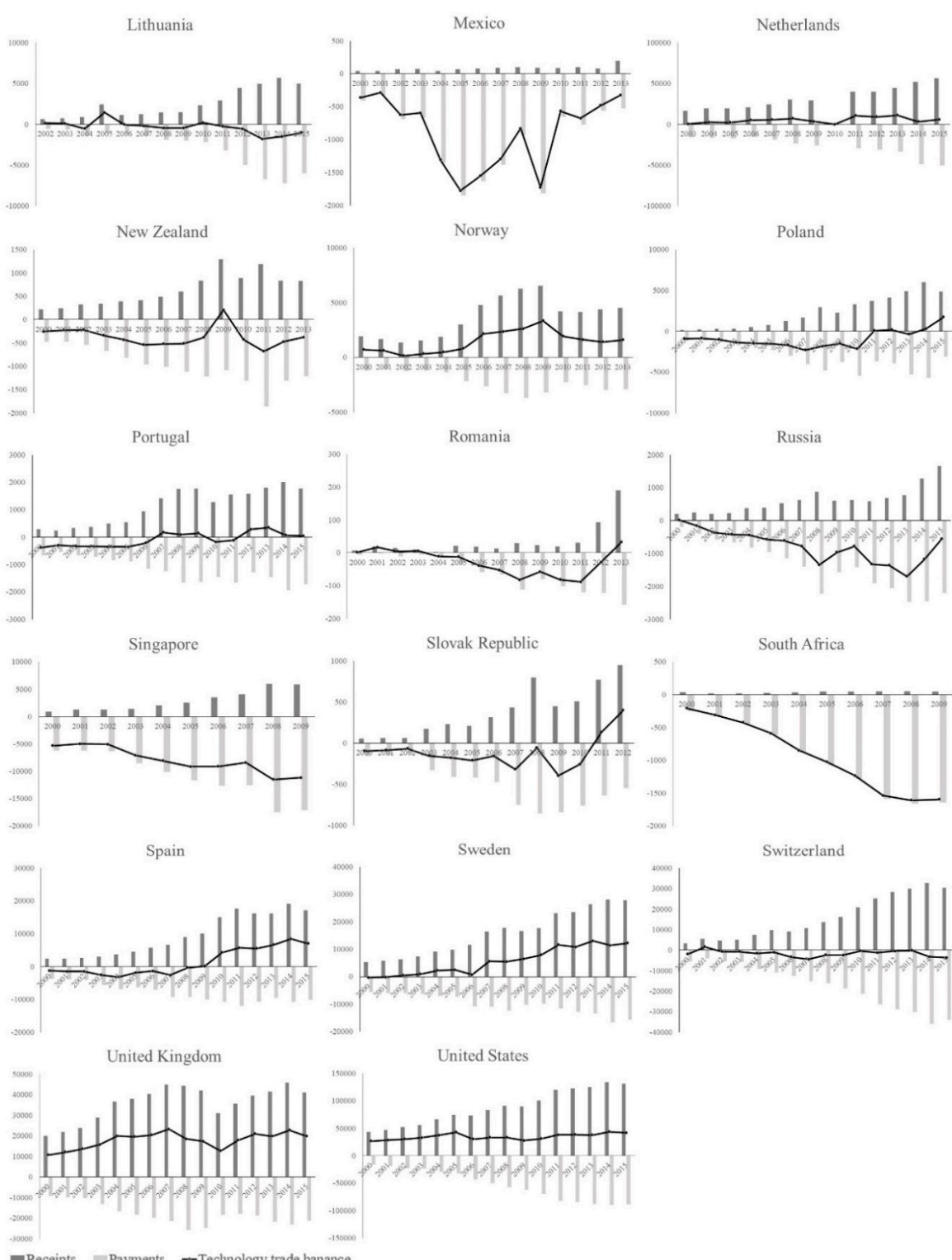

**Figure 2.** Technology balance of payments (TBP) by country. Source: OECD MSTI DB. Notes: *y*-axis (millions of dollars), *x*-axis (year).

### 4.3. Models

Our dataset is a panel. Depending on the treatment of unobserved heterogeneity, we can adopt either fixed effects or random effects regressions. A Hausman test provides the determination between fixed effects and random effects. The result shows that fixed effects regression is preferable over

random effects for all models (Prob > chi2 = 0.000). In addition, fixed effects regressions can eliminate time-invariant unobserved firm heterogeneity, explaining the longitudinal change. Thus, fixed effects regressions are considered "extremely stringent tests" [70]. Recognizing this, we employ a fixed-effect regression model for analysis with fixed year effects. The employed model is:

$$\text{Inward FDI}_{t+1} = \alpha_{0i} + \lambda_t + \alpha_1\,\text{TCHC}_{i,t} + \alpha_2\,\text{institutional quality}_{i,t} +$$

$$\alpha_3\,\text{TCHC} \times \text{institutional quality}_{i,t} + \alpha_4\,\text{Controls}_{i,t} + e_{i,t}$$

where $\alpha_0$ refers to firm fixed effects, $\lambda_t$ represents.

To check the robustness of the findings, we also test generalized estimating equations (GEEs) to account for any within-subject correlations, avoiding spurious results from first-order autoregressive correlations. The method also provides robust variance estimates, account for heteroscedasticity and unobserved differences among firms [71]. The results are consistent with what we have from a fixed-effect regression.

## 5. Results

Table 1 presents descriptive statistics that include means, standard deviations and correlations among all the variables. Table 2 presents the results of fixed effects regressions to test the relationship between the technological capability of a host country (TCHC) and inward FDI. Model 2 shows that TCHC has an association (curvilinear U-shaped) with inward FDI, indicating that both low and high TCHC are associated with a higher inward FDI (TCHC: $\beta = -1932.91$, $p < 0.1$; TCHC$^2$: $\beta = 148.90$, $p < 0.001$). From Model 3 to Model 7, we include each institutional quality variable and test its main effect. While institutional quality variables show no statistically significant effects, the results of a U-shaped relationship between TCHC and inward FDI remains unchanged. The findings in Table 2 provide support for H1.

**Table 1.** Descriptive statistics [a].

| Variable | Obs. | Mean | S.D. | Min | Max | 1 | 2 | 3 | 4 | 5 | 6 | 7 |
|---|---|---|---|---|---|---|---|---|---|---|---|---|
| 1. Inward FDI [c] | 489 | 2.44 [b] | 7.14 [b] | −2.49 [b] | 73.4 [b] | | | | | | | |
| 2. TCHC [d] | 488 | 2.59 | 8.29 | −24,754 | 43.87 | 0.56 | | | | | | |
| 3. Government size | 489 | 6.20 | 0.91 | 3.77 | 8.37 | 0.05 | 0.05 | | | | | |
| 4. Regulation | 489 | 7.46 | 0.82 | 5.06 | 9.11 | 0.23 | 0.27 | 0.10 | | | | |
| 5. Sound money accessibility | 489 | 9.23 | 0.87 | 2.71 | 9.89 | 0.08 | 0.16 | −0.16 | 0.38 | | | |
| 6. Property right | 489 | 7.12 | 1.23 | 3.61 | 9.14 | 0.12 | 0.15 | −0.21 | 0.59 | 0.51 | | |
| 7. Total GDP | 489 | 12.97 | 1.42 | 9.20 | 16.72 | 0.41 | 0.58 | 0.10 | 0.15 | 0.21 | 0.14 | |
| 8. Commodity trade (net) | 465 | 1.16 | 6.94 | −16.83 | 47.62 | −0.07 | −0.06 | −0.038 | 0.13 | 0.01 | 0.02 | 0.05 |

[a] All correlations with magnitude |0.08| are significant at the 0.05 level. [b] = ($\times 10^7$). [c] The median is 67.02 in the sample. [d] TCHC = technological capability of a host country.

Table 3 illustrates the moderating effects of institutional quality on the relationship between TCHC and inward FDI. In Hypothesis 2, we argue that *government size* will positively moderate the relationship between TCHC and inward FDI, indicating steeper slopes on both sides of the baseline U-shape. This argument is supported by the result of Model 1 ($\beta = 1.62$, $p < 0.01$). For Hypothesis 3, we predict that the *regulation* system will positively moderate the relationship between TCHC and inward FDI, indicating steeper slopes on both sides of the baseline U-shape. Model 2 provides support for this hypothesis ($\beta = 1.73$, $p < 0.05$). Hypothesis 4 proposes that *sound money accessibility* will negatively moderate the relationship between TCHC and inward FDI, implying flatter slopes on both sides of the baseline U-shape. The results of Model 3 support this hypothesis ($\beta = -6.56$, $p < 0.001$). In Hypothesis 5, we posit that *property rights* will negatively moderate the relationship between TCHC and FDI, meaning that both sides of the baseline U-shape become less steep. Model 4 provides support

for this argument ($\beta = -1.58$, $p < 0.01$). Becoming steeper also implies that a unit change in FDI becomes more sensitive to a unit change in TCHC while being less steep explains the opposite case.

**Table 2.** Fixed-effect regression on the relationship between technological capability of a host country (TCHC) and foreign direct investment (FDI).

| | FDI $_{t+1}$ | | | | | | |
|---|---|---|---|---|---|---|---|
| **Variables** $_t$ | **Model 1** | **Model 2** | **Model 3** | **Model 4** | **Model 5** | **Model 6** | **Model 7** |
| TCHC | 1.50 * | −1932.91 [+] | −1.97 * | −2.01 * | −1.93 [+] | −1.75 [+] | −1.85 [+] |
| | (0.76) | (999.95) | (1.00) | (1.01) | (1.00) | (1.00) | (1.02) |
| TCHC$^2$ | | 148.90 *** | 0.15 *** | 0.15 *** | 0.15 *** | 0.14 *** | 0.14 *** |
| | | (29.38) | (0.29) | (0.03) | (0.03) | (0.03) | (0.03) |
| Government size | | | 4.03 | | | | 3.62 |
| | | | (6.27) | | | | (6.47) |
| Regulation | | | | 3.08 | | | 3.21 |
| | | | | (6.67) | | | (6.80) |
| Sound money accessibility | | | | | −0.94 | | −0.61 |
| | | | | | (5.18) | | (−13.7) |
| Property right | | | | | | −13.32 | 11.6 |
| | | | | | | (8.33) | (8.38) |
| Total GDP [b] | 10.98 | 9876.60 | 7.86 | 8.43 | 11.47 | 13.78 | 11.6 |
| | (13.30) | (12,889.4) | (13.28) | (13.28) | (15.61) | (13.09) | (16.5) |
| Commodity trade (net) | −0.02 | −113.19 | −0.11 | −0.10 | −0.12 | 0.39 | −0.14 |
| | (0.40) | (388.99) | (0.39) | (0.39) | (0.39) | (3.39) | (0.39) |
| R$^2$ | 0.31 | 0.37 | 0.35 | 0.38 | 0.36 | 0.27 | 0.27 |
| N | 431 | 431 | 431 | 431 | 431 | 431 | 431 |
| F-statistic | 2.10 ** | 3.54 *** | 3.37 *** | 3.36 *** | 3.35 *** | 3.5 *** | 3.03 *** |
| DoF | 381 | 380 | 379 | 379 | 379 | 379 | 376 |

[a] Year dummy is included, but not reported here. [b] Natural logarithm. [c] All coefficients and standard errors are scaled by $\times 10^6$ to make them presentable in table. [+] $p < 0.10$; * $p < 0.05$; ** $p < 0.01$; *** $p < 0.001$.

**Table 3.** Fixed-effect regression: moderating effects of institutional quality.

| | FDI $_{t+1}$ | | | |
|---|---|---|---|---|
| **Variables** $_t$ | **Model 1** | **Model 2** | **Model 3** | **Model 4** |
| TCHC | −11.49 ** | −14.54 * | 61.86 *** | 10.76 * |
| | (3.65) | (6.03) | (12.20) | (4.82) |
| TCHC$^2$ | 0.129 *** | 0.11 ** | 0.12 *** | 0.11 *** |
| | (0.030) | (0.04) | (0.03) | (0.03) |
| Government size | 1.77 | | | |
| | (6.28) | | | |
| TCHC x Gov't size | 1.62 ** | | | |
| | (0.60) | | | |
| Regulation | | −0.70 | | |
| | | (6.87) | | |
| TCHC x Regulation | | 1.73 * | | |
| | | (0.82) | | |
| Sound money accessibility | | | 1.99 | |
| | | | (5.04) | |
| TCHC x Sound money accessibility | | | −6.56 *** | |
| | | | (1.25) | |
| Property right | | V | | −10.14 |
| | | | | (8.35) |
| TCHC x Property right | | | | −1.58 ** |
| | | | | (0.60) |

**Table 3.** *Cont.*

| Variables t | FDI t+1 | | | |
|---|---|---|---|---|
| | **Model 1** | **Model 2** | **Model 3** | **Model 4** |
| Total GDP [b] | 5.57 | 7.16 | 6.28 | 17.71 |
| | (13.19) | (13.23) | (15.12) | (13.07) |
| Commo trade(net) | −0.45 | −0.04 | −0.03 | −0.15 |
| | (0.37) | (0.39) | (0.38) | (0.39) |
| $R^2$ | 0.37 | 0.40 | 0.40 | 0.28 |
| N | 431 | 431 | 431 | 431 |
| F-statistic | 3.62 *** | 3.44 *** | 4.78 *** | 3.73 *** |
| DoF | 378 | 378 | 378 | 378 |

[a] Year dummy is included, but not reported here. [b] National logarithm. [c] All coefficients and standard errors are scaled by $\times 10^6$ to make them presentable in table. * $p < 0.05$; ** $p < 0.01$; *** $p < 0.001$.

## 6. Discussion and Conclusions

### 6.1. Academic Contributions

There have been many studies focusing on the determinants of FDI. Among them, some scholars have focused on the impact of FDI on international technology spillovers [6–8]. To demonstrate contingencies between the technological capabilities of a host country and institutional qualities, this study employs the moderation effects between them on inward FDI. The reasons for understanding the dynamic aspects of countries' technological capabilities are critical: (1) technological capabilities evolve continuously, such as South Korea [72]; although rare, the improvement changes the nature of FDI. (2) FDI is bilateral. In many cases, FDI occurs from developed to developing countries. However, there are cases of opposite FDI direction, i.e., from developing to developed. Thus, understanding the motives of FDI based on its goal is critical. In addition, institutional quality matters. Prior research has primarily focused on the direct relationship between institutional quality and FDI [2–10]. However, we posit that institutional quality works as a secondary determinant. We focus on the role of technology capabilities of the host country on inward FDI, with moderations by several institutional quality measures.

The results are interesting. First, there is a curvilinear relationship between the technological capability of the host country and inward FDI, which means that the FDI is especially high when countries have either extremely high technological capability or extremely low technological capability.

While previous studies have maintained the linear perspective on the relationship between technological assets and FDI [11–14] in both developed and developing economies contexts, our study offer an alternative perspective that indicates the inward FDI may vary in different level of technological capability of the host country. It makes sense from the perspective of opportunity exploration by investing in foreign companies. When investing companies explore new markets, they want to avoid fierce competition with local companies. Foreign companies naturally have disadvantages, called liability of foreignness. They can only sustain their business when the product or service has a comparative advantage relative to those by local companies. Similarly, investing companies have an evident goal to access technological capabilities when they decide to allocate their resources to countries with higher technological capability. Second, this trend can be considered together with the type of FDI. When considering resource-seeking FDI and efficiency-seeking FDI, investors would prefer countries specialized in technology import in terms of having technologies for securing and utilizing resources and technologies for production for off-shoring activities, respectively. On the other hand, investment companies pursuing knowledge-seeking FDI to develop innovative technologies through capital investment would prefer countries with technological capabilities to specialize in technology exports. All these explanations show the effectiveness of the internalization advantages of the recipient country, which are one of the determinants of FDI.

Based on this main finding, we explore the diverse contingencies by moderating institutional quality measures: size of government, legal structure and security of property rights, access to sound money and regulation of credit, labor and business (from the Fraser Institute). The results tell interesting stories. Although the four institutional variables did not show direct significance, there were significant results in terms of moderation effect. The negative result for legal structure and security of property rights and access to sound money signifies a reduction in the acceleration effect of FDI by specialization of export or import of technology trade. Both institutional variables are about the degree to which the value of the investment is preserved, and the soundness of the currency is maintained. Evidently, even if the specialization of technology imports or exports is increased, the positive effect on FDI is not further accelerated. On the other hand, in the case of the size of government and legal structure and security of property rights, the positive (+) effect of FDI accelerated by specializing in technology trade imports or exports was shown; rather than securing the asset value, the scale of or regulation by the government enhances the specialized effect of technology trade as an investment determinant for FDI. While previous studies have mainly to investigate the direct and linear association between institutional quality and FDI ([5–8,10]), based on the U-shaped relationship between the technological capability of the host country and inward FDI, our study illustrates that institutional quality can moderate this relationship, implying that it may accelerate (or alleviate) both negative and positive relationship between the technological capability of the host country and inward FDI when the degree of technological capability of the host country varies.

In summary, this study tried to show factors affecting FDI inflows from the viewpoint of technical competence and institutional variables that were previously carried out individually through one analysis model. First, the existing studies related to technology capabilities have discussed two directions: one is that the low technology level in developing countries is improved by advanced countries' FDI [11–15]. The other is that FDI investment occurs between high technology level countries for the purpose of technology development or knowledge accumulation [13,14,22]. Both of these aspects could be seen as a single analysis model through the U-shaped curve. Second, in the case of the institutional quality, the aspect of the "rules of the game" itself [33–35] was empirically shown through the moderation effect. The positive moderation effect [34,55–58] and the negative moderation effect [58,59] were identified using four institutional variables. Through this study, the possibility of utilizing the TBP variable that can represent the technological capabilities was identified, and the extended FDI analysis model in which both the institutional and technological capability were considered was explored.

## 6.2. Policy Implications

The preceding conclusions lead to the following policy implications for the countries wishing for FDI. First, to induce FDI, it is necessary to have a strategy that specializes in technical competence appropriate to the current situation of the country. In other words, in the case of countries that have strengths in resources or labor that could increase the efficiency of production, the establishment of a technology imports friendly environment and specialization in technology imports are necessary, with the deficit in technology trade made up for by exporting products. East Asian countries such as Korea and Taiwan have made effective use of this strategy in the past. Second, in the case of technology export-specialized countries with a world-class level of technology, it is necessary to attract various joint technology development research by reorganizing the government's spending activities and regulatory systems to allow fair competition by foreign FDI investors. Such strategies could be useful when a country with a deficit in technology trade seeks to convert to a technology export-specialized one.

As a future study, we propose a case study that analyzes the types of FDI invested in relevant countries and their performance, focusing on both the countries specializing in technology exports and those specializing in technology imports.

### 6.3. Limitation and Future Research Opportunities

The FDI data used in this study include all the various purpose-type FDIs; as a future study, it would be possible to explore more detailed correlations by classifying FDI by type based on the results of this study. Specifically, it would be useful to analyze the determinants of FDI by type, changing trends in FDI types, and the causal relationship between FDI types and economic growth in terms of the growth strategy and policy establishment of the investee countries.

The TBP of each country varies from year to year, and during a certain time period, specific countries have very low or very high values compared to countries with a similar technology level. The limitations of this study can be supplemented by conducting country-by-country analysis to reflect such country specificity and carrying out country case reviews to identify the causes of such phenomena.

The other avenue for future research can be investigating the association between the technological capability of the host country and different types of FDI. Existing literature (e.g., [73–75]) classifies FDI mode into four categories such as market-seeking, efficiency-, Resource-seeking and strategic asset-seeking. For instance, one can compare the patterns of TBP between a country that largely focuses on market-seeking and a country that mainly received strategic asset-seeking FDI. Alternatively, exploring the types of technologies based on different modes of FDI could provide an interesting angle to explore this area.

**Author Contributions:** All sections including conceptualization, methodology, formal analysis, writing—original draft preparation, review and editing S.K. and B.C. All authors have read and agreed to the published version of the manuscript.

**Funding:** Byungchul Choi's work was supported by the Hankuk University of Foreign Studies Research Fund of 2020.

**Conflicts of Interest:** The authors declare no conflict of interest.

## Appendix A

| | |
|---|---|
| Size of Government: Expenditures, Taxes, and Enterprises (https://www.fraserinstitute.org/studies/economic-freedom) | |
| A | General government consumption spending |
| B | Transfers and subsidies as a percentage of GDP |
| C | Government enterprises and investment |
| D | Top marginal tax rate |
| Legal Structure and Security of Property Rights | |
| A | Judicial independence (GCR) |
| B | Impartial courts (GCR) |
| C | Protection of property rights (GCR) |
| D | Military interference in rule of law and the political process (CRG) |
| E | Integrity of the legal system (CRG) |
| F | Legal enforcement of contracts (DB) |
| G | Regulatory restrictions on the sale of real property (DB) |
| Access to Sound Money | |
| A | Money growth |
| B | Standard deviation of inflation |
| C | Inflation: Most recent year |
| D | Freedom to own foreign currency bank accounts |
| Regulation of Credit, Labor, and Business | |
| A | Credit market regulations |
| B | Labor market regulations |
| C | Business regulations |

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
