# Peer review of "The Impact of the Technological Capability of a Host Country on Inward FDI in OECD Countries: The Moderating Roles of Institutional Quality"

_sustainability, doi:10.3390/su12229711_

Round 1
Reviewer 1 Report
- Is TBP a credible measure of a country's technology capability? While Japan and Germany do have a persistent TBP surplus and they are known to be technologically strong, the data for some other countries in Figure 2 are far less convincing. For example, compare France, Italy, and Israel. France and Italy display negative or nearly zero TBP while Israel's TBP is persistently positive. So it follows that Israel is technologically more capable than France and Italy. Do anybody believe that? Russia is another incredible case. it runs a huge and persistent TBP deficit and so it is technologically weak compared with Spain or Portugal which run a small TBP surplus. Do anybody believe that?
- Line 80 - 81 "Previous empirical studies have focused on the existence and variance of FDI". What do you mean? The FDI literature is huge and there are many topics, but I am not sure if existence and variance have been well studied. The ensuring sentences say something about FDI spillover which is indeed a big topic in the literature, but I am not sure the logical connection between these sentences and the "Previous empirical studies ..." sentence.
- Line 86-87 "only few studies have investigated the effect host country's technological capabilities on FDI." This is not true. The FDI literature uses the term "absorptive capacity" rather than "technological capability". Simply google search "absorptive capacity and FDI" and you can find many recent papers.
- Incomprehensible sentences in Line 128-132 "From the perspective of investees ... FDI is beneficial: MNE deploys their investment to access the assets of developing countries; these assets include technology, ...The spillovers of such capabilities are crucial assets ...". The problem is the contents, not the English language.
- Section 3.1. The arguments for the U-shaped relationship between FDI and technology capability is unconvincing. One can tell a story and argue the other way round or whatever shape. There is no model whatsoever in the paper. Line 53 "This study hopes to theoretically and empirically contribute to FDI related research". I don't see where the paper makes theoretical contribution to the literature.
- Line 19-20 in the abstract "Our study contributes to the literature on FDI and institutional quality in the context of innovation". The "in the context of innovation" part seems to be an over-statement. I don't see any relationship between the findings in the paper and innovation.
- Line 258-259 "We draw on diverse institutional qualities of each country in the sample employed for the Granger causality analysis". Where is the Granger causality test or analysis in the paper?
- Line 259-260 "Our final sample comprises 249 country-year observations ...", but the sample size N reported in Table 2 is 431. Why is there inconsistency?
- Table 2, footnote b "Logarithm". What is in logarithm? Presumably some of the variables in the regression are measured in log. The author should make it clear.
- Table 2, the "F" on the last line. Presumably it is some sort of F-statistic. What are the degrees of freedom? What is the null hypothesis being tested?
- Line 407-409 "Rather than focus on the static condition of FDI flow, this study highlights the dynamic aspects of technological capabilities..." Again, this is an over-statement. I just don't see any static vs dynamic elements in the empirical work.
Author Response
Dear reviewer,
We would like to thank you for giving us the opportunity to revise and resubmit this manuscript; it has been revised according to your recommendations and has certainly benefited from a number of your detailed suggestions. Thank you again for your useful comments, corrections and suggestions. We have responded sequentially to each suggestion below. (All modification and newly added paragraphs are highlighted in blue for visual recognition)

Reviewer 2 Report
This paper is well written and addresses an interesting subject.Results are obtained from a fixed effects panel estimation. I have few questions about the methodology applied:
1) why the panel is estimated with fixed effects rather than random effects?
2) Since it is largely confirmed that FDI flows are more likely to be driven by the economic conditions in the host countries, it is natural to expect that there is an endogeneity problem between FDI and the istitutional quality. Authors use lagged values of control variables to overcome this problem, however the endogeneity (and the way to treat it) should be better discussed. Moreover I suggest to use a GMM estimator or, at least, to provide a robustness check.
Author Response
We would like to thank you for giving us the opportunity to revise and resubmit this manuscript; it has been revised according to your recommendations and has certainly benefited from a number of your detailed suggestions. Thank you again for your useful comments, corrections and suggestions. We have responded sequentially to each suggestion below. (All modification and newly added paragraphs are highlighted in blue for visual recognition)

Reviewer 3 Report
The paper is interesting as it explores the link between between the technology capability of a nation, institutional quality, and inward FDI. Based on fixed-effects regression, the reserach results show that there is a U-shape relationship between net technology capability of a host country and inward FDI.
The following suggestions are made to authors:
In the Abstract, the first sentence: "This study explores the link between, on the one hand, the technology capability and institutional quality of a host country, and, on the other, inward foreign direct investment (FDI)" is not clear, and it is advised to reformulate it.
In the last part of the paper, authors discuss the results, but they do not refer to any previous research to show are those similar or different from their research. It is suggested to authors to add some examples of other research results as it will put their results in a wider perspective. Also, authors conclude that the FDI is especially high when countries have either extremely high technology capability or extremely low technology capability. They explain that it makes sense from the perspective of opportunity exploration by investing foreign companies. However, they do not distinguish different types of FDI in their research (e.g. resource seeking from market seeking FDI, or sectoral division of FDI). It is suggested that authors also comment on the country results in order to better explain their conclusion (i.e. to compare the results in the high technology capability and extremely low technology capability countries).
Author Response
We would like to thank you for giving us the opportunity to revise and resubmit this manuscript; it has been revised according to your recommendations and has certainly benefited from a number of your detailed suggestions. Thank you again for your useful comments, corrections and suggestions. We have responded sequentially to each suggestion of you based on point-by-point. (All modification and newly added paragraphs are highlighted in blue for visual recognition).
